# Investigating the Antimicrobial Activity of Anuran Toxins

**DOI:** 10.3390/microorganisms13071610

**Published:** 2025-07-08

**Authors:** Manuela B. Pucca, Anne Grace A. C. Marques, Ana Flávia M. Pereira, Guilherme Melo-dos-Santos, Felipe A. Cerni, Beatriz C. S. Jacob, Isabela G. Ferreira, Rafael L. Piccolo, Marco A. Sartim, Wuelton M. Monteiro, Isadora S. Oliveira

**Affiliations:** 1Department of Clinical Analysis, School of Pharmaceutical Sciences, São Paulo State University (UNESP), Araraquara 14800-903, SP, Brazil; rl.piccolo@unesp.br; 2Post Graduate Program in Tropical Medicine (PPGMT), Amazonas State University, Manaus 69065-001, AM, Brazil; annegracecunha@hotmail.com (A.G.A.C.M.); marcosartim@hotmail.com (M.A.S.); wueltonmm@gmail.com (W.M.M.); 3Graduate Program in Bioscience and Biotechonology Applied to Pharmacy, School of Pharmaceutical Sciences, São Paulo State University (UNESP), Araraquara 19060-900, SP, Brazil; guilherme.melo-santos@unesp.br; 4The Center for the Study of Venoms and Venomous Animals (CEVAP), São Paulo State University (UNESP), Botucatu 18619-002, SP, Brazil; ana.f.pereira@unesp.br; 5Medical School, Federal University of Roraima (UFRR), Boa Vista 69310-000, RR, Brazil; felipe_cerni@hotmail.com; 6Department of BioMolecular Sciences, School of Pharmaceutical Sciences of Ribeirão Preto, University of São Paulo, Ribeirão Preto 19040-903, SP, Brazil; beatrizjacobcs@gmail.com (B.C.S.J.); igobboferreira@yahoo.com.br (I.G.F.); 7School of Pharmaceutical Sciences, Federal University of Amazonas, Manaus 69077-000, AM, Brazil

**Keywords:** toad, frog, antimicrobial peptides, alkaloid

## Abstract

Anurans, commonly known as frogs and toads, comprise a diverse group of amphibians distributed across all continents except Antarctica. This manuscript provides a detailed overview of the global anuran fauna, emphasizing their biology, remarkable adaptations, and ecological importance. A particular focus is placed on their specialized cutaneous glands, which are crucial for defense, communication, and survival. These glands secrete a diverse array of bioactive compounds, including peptides, alkaloids, and other secondary metabolites, shaped by evolutionary pressures. Among these compounds, toxins with potent antimicrobial properties stand out due to their ability to combat a broad spectrum of microbial pathogens. We explore the chemical diversity of these secretions, analyzing their modes of action and their potential applications in combating antibiotic-resistant bacteria and other pathogens. By integrating knowledge, this study underscores the importance of anurans as both ecological keystones and a valuable resource for biotechnological innovations. Furthermore, it highlights the urgent need to conserve anuran biodiversity for harnessing their potential in the development of novel antimicrobial agents to address global health challenges.

## 1. Introduction

Animal venoms and poisons are widely recognized as rich sources of bioactive molecules with diverse pharmacological properties [1]. Among these, anurans—commonly known as frogs and toads—are particularly notable for their unique skin secretions, which serve as a crucial chemical defense mechanism against predators, microbial infections, and environmental stressors [2]. These amphibians inhabit humid and often microbially dense environments, where they are constantly exposed to potential pathogens. But how do these amphibians manage to survive and thrive in such conditions while exhibiting remarkable resistance to bacterial and fungal infections? This intriguing phenomenon suggests that their skin secretions contain potent antimicrobial compounds, making them an important source of study in the bioprospecting for novel therapeutic agents [3,4]. This review aims to provide a comprehensive analysis of the antimicrobial properties of anuran toxins, summarizing key findings from recent studies, highlighting species known for their antimicrobial activity, and exploring the underlying mechanisms by which these compounds exert their effects. By consolidating current knowledge, we seek to underscore the potential of anuran-derived molecules in the development of next-generation antimicrobial agents and encourage further research into this promising yet underutilized field.

## 2. Anuran Fauna

Amphibians of the order Anura are four-legged vertebrate animals (Figure 1). Currently, there are 7664 species distributed in 54 taxonomic families, including toads, frogs, and tree frogs [5,6]. They are found throughout the world, except in the polar regions of Antarctica and Greenland, with the greatest diversity in tropical regions. In general, toads are more commonly found in forested regions, frogs are more prevalent in savannah areas, and tree frogs are most abundant in both forest and Atlantic forest regions [6]. Brazil is one of the richest areas in amphibian species in the world, with around 1200 specimens recorded [6,7].

### 2.1. Toads

Toads are widely distributed across the globe, typically found in tropical regions inhabiting terrestrial/aquatic environments, such as lakes, rivers, and marshes. They are characterized by their smooth, dry skin, the presence of paratoid glands, and short legs. They are generally 5 to 20 cm long, but some species can reach up to 30 cm in length [6].

*Rhinella schneideri*, generally known as the cane toad, is an anuran from the Bufonidae family that has a wide geographic distribution throughout South America, including Brazil, Argentina, Bolivia, Paraguay, and Uruguay. It is the best known of the amphibians, as it appears in backyards and home gardens and can reach up to 25 cm in length. A striking characteristic of the genus *Rhinella* is the presence of well-developed parotid glands, located dorsally close to the tympanum. These glands are responsible for producing a yellowish, dense, very complex, and variable toxic secretion, depending on the species [8,9].

### 2.2. Frogs

They are characterized by their moist and smooth skin, the absence of paratoid glands, and long, muscular legs, with fingers without projections and a robust waist. They are generally 5 to 15 cm long, but some species can reach up to 25 cm in length [6,8]. The bullfrog (*Rana catesbeiana*) is one of the best-known species, as it is widely consumed by humans, promoting the development of aquaculture in several countries [10,11].

### 2.3. Tree Frogs

Tree frogs generally range in length from 2 to 8 cm, but some species can reach up to 10 cm in length. They are adapted to an arboreal life, have a slender waist, generally smooth skin, long and thin legs, as well as adhesive disks on the tips of their digits, allowing them to climb vertical surfaces. They have paratoid glands located in different parts of the body, depending on the species [6,8].

The most studied genus is *Phyllomedusa*, for example, *Phyllomedusa bicolor* from Amazonian forests. Popularly known as “Kambô”, “Kampô”, or “Kampu”, the natives, mainly from Brazil, use the poison secreted by the skin of this animal in traditional medicine, called “frog vaccine”, as a ritual to prevent diseases, and it is considered a physical and mental invigorator [12,13].

## 3. Secretions from Anuran

Anurans possess two main types of skin glands: mucous glands, associated with moisture maintenance and cutaneous respiration, and granular glands, specialized in secreting chemical substances for defense [14,15]. This duality reflects the direct exposure of their skin to environmental challenges such as desiccation, pathogens, and predators [16]. Thus, anuran skin has evolved to produce a diversity of bioactive compounds, including alkaloids, biogenic amines, steroids, peptides, and proteins, making them valuable models for chemical and biological research [16,17]. The composition of these secretions varies significantly among families and species and will be explored in this review, regarding their antimicrobial ability.

Among the most notable bioactive components of anuran cutaneous secretions are alkaloids, whose discovery and characterization have advanced significantly due to improvements in analytical techniques. Currently, over 800 distinct alkaloids have been identified and characterized in anuran secretions, with new representatives being described annually, expanding our knowledge of this fascinating chemical diversity [18]. These alkaloids can be classified into three main structural groups—guanidinics, lipophilics, and indolics—which present distinct chemical and physiological properties [19]. Some alkaloids deserve special attention for their remarkable pharmacological potential. Concentrated mainly in the Bufonidae family, these compounds have demonstrated in experimental studies muscle-modulating activities, as well as promising anticancer, antiviral, and antimicrobial effects [19]. The structural diversity of these indole alkaloids makes them particularly interesting for investigations in medicinal chemistry and new drug development.

Currently, over 2000 bioactive peptides have been identified in anuran secretions, classified into about 100 distinct families [16]. Present in all species studied, these peptides exhibit remarkable structural diversity that reflects specific adaptations to the different ecological niches occupied by these animals. This molecular variability is directly related to the particular selective pressures of each habitat, including interactions with pathogens, predators, and environmental factors that have driven their molecular evolution [14,16]. Among them, antimicrobial peptides stand out not only for being the most abundant but also for their broad spectrum of activity, acting as a first line of defense against bacteria, fungi, and viruses, and beyond their antimicrobial properties, many of these peptides present multifunctional properties, including neurotoxic, cardiotoxic and myotoxic effects against potential predators, as well as modulatory activities such as analgesic action and immunoregulation [16,20,21].

## 4. Antimicrobial Compounds

Antimicrobial peptides (AMPs) are commonly found in animal toxins, typically displaying a cationic nature and comprising a small number of amino acids (10 to 50). These peptides tend to adopt an amphipathic α-helix conformation, enabling them to interact effectively with membranes. AMPs showcase a broad spectrum of activity, targeting bacteria, fungi, viruses, and protozoa [22,23]. Within anurans, there exists a diverse array of AMPs that play a crucial role in their innate immune defense. These peptides are synthesized and secreted mainly in the skin, acting as a first line of protection against pathogens through their cytolytic activity. Adapted to various ecological niches, different anuran species have evolved distinct AMPs, serving not only as effective defenses against a wide range of microbial invaders but also reflecting adaptations to their unique habitats [24,25]. Thus, anuran toxins can be applied to combat different microorganisms, which is shown in Figure 2.

There are models that explain the actions of AMPs on cell membranes, with the most well-known being the barrel-stave model, toroidal model, and carpet-like model. In the barrel-stave model, peptides form a barrel-shaped pore; the hydrophobic residues of the peptides interact with the lipid part of the membrane, while the hydrophilic residues form the interior of the pore, allowing the release of cytoplasmic contents and leading to cell death [26]. In the toroidal model, peptides align with the phospholipids of the membrane and fold to form hydrophilic pores, enabling membrane permeabilization and disintegration. Finally, in the carpet-like model, peptides accumulate in large amounts and associate parallel to the membrane, creating tension that leads to membrane disruption and cell lysis [25]. In addition, AMPs may also exert intracellular activity, targeting nucleic acids, enzymatic activity, and protein synthesis, among other essential cellular functions [26].

### 4.1. Bacteria

AMPs derived from various anuran families, including Alytidae, Ranidae, Pipidae, Hylidae, Leptodactylidae, Leiopelmatidae, Hyperoliidae, Bombinatoridae, Dicroglossidae, and Myobatrachidae, exhibit potent activity against both Gram-positive and Gram-negative bacteria, commonly tested with strains such as *Staphylococcus aureus* (*S. aureus*) and *Escherichia coli* (*E. coli*) [24,25]. Antibacterial activity has been identified in the skin secretions of 78 species of anurans, yielding a total of 323 antimicrobial peptides with antibacterial activity. Detailed information can be found in Table 1.

In the Alytidae family, the skin secretions of the anurans *Alytes maurus* and *Alytes obstetricans* possess the AMPs alyteserin-1 and alysteserin-2, which have demonstrated effectiveness against *S. aureus* and *E. coli*. The peptides from *Ascaphus truei*, known as ascaphins 1 to 8 and belonging to the Leiopelmatidae family, exhibit antibacterial activity against *S. aureus*, as well as Gram-negative bacterias such as *E. coli*, *Enterobacter cloacae* and *Klebsiella pneumoniae* [32].

The Ranidae family of anurans, specifically within the genera *Amolops*, *Hylarana*, *Lithobates*, *Odorrana*, *Pelophylax* and *Rana*, express a range of AMPs with broad spectrum of action, including brevinin-1, brevinin-1, esculetin-1, esculetin-2, palustrin-1, palustrin-2, ranacyclin, ranatuerin, and temporin [25,29,30,31,40,41,42,56,66,67,68,69,88,89]. Notably, *Rana japonica* features japonicin-1 and japonicin-2, while *Rana nigromaculata* exhibits nigrocin-1 and nigrocin-2 [92,93].

Furthemore, the genus *Litoria* within the Ranidae family exhibits the peptides aurein, citropin, caerin, uperin, and fallaxidin [57,58,59,60,61,62,63]. Another notable AMP is melittin, derived from the bee venom of *Apis mellifera*, showcasing efficacy against both Gram-positive and Gram-negative bacteria [119,120]. Within anuran AMPs, melittin-related peptides (MRPs) found in the Ranidae family emulate melittin, demonstrating a broad-spectrum antimicrobial effect against bacteria [25,102].

Bombinin-related peptides (BLPs) from the Bombinatoridae family, specifically bombinins H, exhibit diminished bactericidal efficacy while demonstrating hemolytic activity on erythrocytes [121]. In contrast, other BLPs, including maximins 1 to 5 from *Bombina maxima*, display robust antibacterial activity [34].

The Dicroglossidae family, represented by *Fejervarya cancrivora*, possesses the tigerinin peptide, similar to *Rana tigerina*, demonstrating antibacterial activity against *S. aureus*, *Bacillus subtilis*, *Pseudomonas aeruginosa*, and *E. coli* [37,106]. In the Hylidae family, found within the genera *Hyla*, *Hylomantis*, *Phyllomedusa*, and *Hypsiboas*, various naturally occurring peptides such as dermaseptins, dermatoxins, distinctin, hylain, hylin, hylaseptin, phylloseptins, phylloxin, and raniseptin [38,39,43,44,45,71,72,73,74].

Kassinatuerins from the Hyperoliidae family are found in the skin secretion of *Kassina senegalensis* [48,49] and *Kassina maculata* [46]. Peptides within the Leptodactylidae family, such as fallaxin (*Leptodactylus fallax*), laticeptin (*Leptodactylus laticeps*), pentadactylin (*Leptodactylus pentadactylus*), and ocellatin (*Leptodactylus ocellatus*) exhibit action majority for Gram-negative bacteria [50,51,52,53,54]. Addiyionally, uperin from *Uperoleia mjobergii* (Myobatrachidae) demonstrates efficacy against a great variety of microrganisms *S. aureus*, *Staphylococcus epidermidis*, *Bacillus cereus*, *Micrococcus luteus*, *Streptococcus uberis*, *Leuconostoc lactis*, *Pasteurella multocida*, and *Listeria innocua.*

The skin secretion of the genus *Xenopus* (*Xenopus amieti*, *Xenopus andrei*, *Xenopus borealis*, *Xenopus clivii*, *Xenopus laevis*, and *Xenopus muelleri*) within the Pipidae family reveals AMPs othologous to the magainins, peptides orthologous to peptide glycine-leucine-amide (PGLa), homologous to caerulein-precursor fragments (CPF), and xenopsin precursor fragment (XPF) [111,112,113,114,115,116,117]. Additionally, the XT 1 to 7 peptides from *Xenopus tropicalis* exhibit some similarities to *X. laevis* PGLa, procaeruleins, and proxenopsin regions [118].

### 4.2. Fungus

Frog skin secretions contain potent compounds that exhibit remarkable activity against various fungi, showcasing their natural defense mechanism in combating fungal infections (Table 2). Among these components, peptides are stored in a granular gland located mainly in the skin of the dorsal region. Three compounds (Arenobufagin, Gamabufotalin, and Telocinobufagin) are described as bufadienolides from boreal toad (*Anaxyrus boreas*) presenting antifungal activities against *Batrachochytrium dendrobatidis* (Bd) being the Arenobufagin the most effective one since it had the lowest estimated concentration where Bd was maximally inhibited (12.9 μg/mL) [122].

Anurans endemic in Indonesia were also studied. One is the bleeding toad *Leptophryne cruentata* and the javan tree frog *Rhacophorus margaritifer* (also named *R. javanus*) and their skin secretions demonstrated antifungal activity against the fungus *Trichophyton mentagrophytes* [123].

Skin secretions of the toad *Bufo arenarum* demonstrated being a rich source of several components with antifungal activity. One of those is an alkaloid called dehydrobufotenine, which demonstrated being effective against phytopathogenic fungi that affect plants of economic interest. After that, forty-five analogs of dehydrobufotenine were synthetized and evaluated for their activity. Six of these analogs demonstrated similar or higher activity than the control used (carbendazim 50 μg/mL). Dehydrobufotenine and most of its analogs showed higher activity against *Alternaria solani* (5–100%) than the controls [124].

*Candida* species are typically harmless fungal microorganisms found in the gastrointestinal and urinary tracts of humans, functioning as commensals. However, under conditions where the immune system is compromised, *Candida* has the capability to transform into a significant cause of severe mucosal or systemic infections, with *Candida albicans* (*C. albicans*). being the most prevalent species [129]. Two peptide components from skin secretion of the European frogs *Pelophylac lessonae/ridibundus*, named Esculentin-1 derived from its N-terminal part, named Esc-1a(1-21)NH_2_ and Esc-1b(1-18)NH_2_ demonstrated being equally active against *C. albicans*. Also, the smallest component (Esc-1b 1-18)NH_2_ demonstrated a dose-dependent membrane-perturbing effect on *Candida* with a kinetic overlapping with killing activity, thus pointing to membrane perturbation as the primary event of its candidacidal activity [125].

Still regarding *C. albicans*, there are reports of several components active against the fungus, such as synthetic peptides (B2RP) from skin secretions of the Southeast Asian frog *Hylarana erythraea* as well as Nigroain and esculetin-2 from the same frog [126,127]. Peptides brevinin-1 and 2 from *H. nigrovittatta* and skin secretion peptide from *H. temporalis* also demonstrated effective activity against the fungus [128].

### 4.3. Virus

Among the AMPs with antiviral activity, those effective against viruses of significant public health relevance stand out, such as human immunodeficiency virus (HIV), herpes simplex virus types 1 and 2 (HSV-1 and HSV-2), Zika virus (ZIKV), dengue virus serotypes 1; 2; 3; and 4 (DENV1-4), severe acute respiratory syndrome coronavirus 2 (SARS-CoV-2), and Influenza A virus subtypes H1N1 and H5N1. Maximin 1 to 5 and Maximin H5, present in *Bombina maxima*, also exhibit antiviral activity against HIV [34].

The genus *Litoria* stands out for its wide variety of antiviral AMPs. Among them are the peptides Caerin 1.2-1.5 and 4.1 (from *Litoria caerulea*), Caerin 1.9 (from *Litoria chloris*), Dahlein 5.6 (from *Litoria dahlii*), Maculatin 1.1 and 1.3 (from *Litoria eucnemis* and *Litoria genimaculata*), Uperin 7.1 (from *Litoria ewingi*), Caerin 1.19 (from *Litoria gracilenta*), Caerin 1.1 and 1.10 (from *Litoria splendida*), and Caerin 1.6 and 1.7 (from *Litoria xanthomera*), all with proven activity against HIV. Additionally, the peptide Frenatin 2, found in *Litoria infrafrenata*, has shown efficacy against the yellow fever virus (YFV) [130,131,132].

A relevant family of peptides is the Dermaseptins, found in *Phyllomedusa sauvagei*. Dermaseptin-S1 acts against HIV, frog virus 3 (FV3), channel catfish virus (CCV), and HSV-1; Dermaseptins-S2 and S3 are effective against HSV-1; Dermaseptin-S4 exhibits anti-ZIKV activity; and Dermaseptin-S9 has demonstrated action against HIV-1 and SARS-CoV-2, standing out as a potential tool in combating the pandemic [133,134,135,136,137,138].

Other notable AMPs include Hylin a1, produced by *Hypsiboas albopunctatus*, which acts against a wide range of viruses, including bovine herpesvirus type 1 (BoHV-1), caprine herpesvirus type 1 (CpHV-1), canine distemper virus (CDV), bovine viral diarrhea virus (BVDV), Schmallenberg virus (SBV), HSV-1, and HSV-2 [139]. The peptide Yodha, found in *Indosylvirana aurantiaca*, has shown efficacy against ZIKV and DENV1-4, the viruses responsible for Zika and dengue diseases [134,140].

In the genus *Rana*, several antimicrobial peptides (AMPs) have been identified, demonstrating antiviral activity against different pathogens. Among them, Esculentin-1ARb, Esculentin-2P, Brevinin-2, Ranatuerin-2P, Ranatuerin-6, and Ranatuerin-9 stand out for their action against HIV [130,141]. Other AMPs, such as Brevinin-1 and Temporin B, are effective against the herpes virus [142,143,144], while Temporin A shows activity against CCV and FV3 [145]. Additionally, Temporin G demonstrates antiviral effects against influenza A virus and human papillomavirus (HPV) [145]. Finally, the peptide AR-23, found in *Rana tagoi*, has shown activity against an impressive variety of viruses, including SARS-CoV-2, measles virus (MeV), human parainfluenza virus type 2 (HPIV-2), human coronavirus 229E (HCoV-229E), BoHV-1, CpHV-1, CDV, BVDV, and SBV [139].

The Magainin family of peptides, found in *Xenopus laevis*, acts against HSV-1 and HSV-2, and Urumin, produced by *Hydrophylax bahuvistara*, has demonstrated efficacy against the H1N1 virus [134,146,147]. The peptide Temporin-SHa, found in *Pelophylax saharica*, also exhibit antiviral activity against HSV-1 [144].

Of the AMPs already isolated and characterized, only 47 exhibit antiviral activity, with proven efficacy against 26 distinct viruses. Table 3 summarizes the detailed information about these molecules, including their species of origin and the viruses they combat. These examples illustrate the potential of AMPs as a new frontier in combating viral infections.

### 4.4. Protozoo

Among the diverse composition of frog skin secretions, several components with anti-parasitic activity have already been identified; this action promoted by AMP’s is being elucidated and has shown promising results against different types of protozoa [152] (Table 4). The mechanism of action responsible for the activity mainly involves the action of these molecules on the permeability of biological membranes, such as Temporins A and B that act on the surface–membrane and also reduce intracellular ATP and dermasipitins responsible for forming amphipathic helices within membranes [153]. In this context, it is possible to highlight how promising the use of these peptides as a treatment for different types of parasitic infections, often neglected but with great potential to cause damage to human health, is promising [154].

## 5. Advanced Research

Advanced research on anuran toxins has driven significant scientific progress, leading to clinical trials, patents, and the development of commercially available products. Toxins such as bufalin have demonstrated remarkable potential in oncological applications [165]. Clinical investigations, including studies on Huachansu for cancer treatment (breast cancer, gallbladder cancer, gastric cancer, hepatocellular carcinoma, liver cancer, lung cancer, non-small cell lung cancer, and pancreatic cancer), are ongoing and highlight the therapeutic promise of these compounds [166].

Currently, there are no antimicrobial products on the market that are directly derived from anuran toxins. However, many of these toxins exhibit promising antimicrobial properties and remain the focus of extensive research. Anurans produce a diverse array of bioactive peptides in their skin secretions, many of which show activity against bacteria, fungi, and other microorganisms. Examples of antimicrobial peptide classes found in anurans have been previously cited in this review.

Although no direct products derived from anuran toxins have reached the market yet, advancements in biotechnology hold significant promise for enabling new therapeutic applications in the future. However, there are currently no reports of clinical trials directly involving anuran toxins or antimicrobial peptides, such as magainins and dermaseptins, for therapeutic use in humans.

Despite this, numerous studies emphasize the potential of these molecules in combating bacterial, fungal, and other infections. Some of these compounds have already undergone preclinical testing in vitro and in vivo to evaluate their efficacy and safety. For instance, dermaseptins exhibit activity against Gram-positive and Gram-negative bacteria, protozoa, and fungi while demonstrating low toxicity to healthy human cells [133].

The interest in advancing these compounds to clinical trials remains strong. Before an antimicrobial molecule candidate can proceed to in vivo studies and subsequently to clinical trials as an alternative therapeutic agent, it must undergo a comprehensive preclinical evaluation. This includes detailed in vitro assays to determine its antimicrobial potency, spectrum of activity, cytotoxicity against cells, and potential to induce resistance. The compound must also be tested for its mechanism of action and synergy with existing antibiotics. Promising candidates are then advanced to in vivo efficacy studies in relevant infection models to assess pharmacokinetic and pharmacodynamic (PK/PD) parameters, bioavailability, tissue distribution, metabolism, and toxicity. If the in vivo results support safety and efficacy, a regulatory submission—such as an Investigational New Drug (IND) application—must be prepared, including all preclinical data. This step is essential to gain approval from regulatory agencies to initiate Phase I clinical trials focused on evaluating safety, tolerability, and preliminary pharmacokinetics in humans. Such a process ensures that only the most viable and safe antimicrobial candidates progress as potential therapeutic alternatives [167,168,169,170]. So, it is a long lasting protocol that hinges on improvements in formulation, delivery methods, and modifications to their pharmacokinetic and pharmacodynamic properties. Current efforts are focused on developing more stable and less toxic analogs to overcome these limitations and unlock their therapeutic potential, such as magainins and synthetic analogs [171]. Department of Health and Human Services.

## 6. Conclusions

In recent years, there has been a growing focus on the antimicrobial potential of anuran-derived toxins, especially peptides and alkaloids, which demonstrate broad-spectrum activity against bacteria, fungi, viruses, and other pathogens. This interest is driven largely by the escalating global crisis of antimicrobial resistance [172,173], which urgently demands novel therapeutic agents beyond conventional antibiotics. Anuran secretions, evolved as chemical defense mechanisms, offer a rich and largely untapped reservoir of bioactive molecules with unique structures and modes of action that differ from traditional antimicrobials, making them highly attractive candidates for drug discovery. Despite the promising nature of these compounds, significant knowledge gaps remain. While a handful of species—primarily from well-studied regions—have been the focus of intensive research, the vast majority of anuran biodiversity, especially in ecologically rich yet underexplored habitats such as the Amazon rainforest, Southeast Asian tropical forests, and African wetlands remains uncharacterized. This presents a compelling frontier for future bioprospecting and biodiversity-driven drug discovery efforts. Moreover, understanding the precise mechanisms by which these peptides and alkaloids exert their antimicrobial effects is still incomplete. Insights into their interactions with microbial membranes, immunomodulatory properties, and potential synergy with existing antibiotics could unlock new therapeutic strategies. Additionally, advances in synthetic biology, peptide engineering, and high-throughput screening are enabling the design and optimization of these natural molecules to enhance their stability, reduce toxicity, and improve pharmacokinetic profiles, accelerating their translation from bench to bedside. For researchers, this emerging field highlights the importance of multidisciplinary approaches combining ecology, chemistry, pharmacology, and computational modeling to fully harness anuran toxins’ therapeutic potential. For clinicians and the drug development field, these molecules represent a promising avenue for addressing antimicrobial resistance through novel mechanisms and chemical diversity that could circumvent existing resistance pathways.

In summary, anuran-derived antimicrobial agents not only expand our arsenal against resistant pathogens but also exemplify the value of biodiversity as a critical resource for future drug innovation. Continued exploration, coupled with technological advancements, will be key to realizing their full biomedical potential.

## Figures and Tables

**Figure 1 microorganisms-13-01610-f001:**
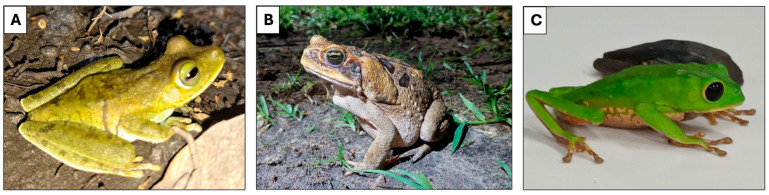
Representative anuran species. (**A**) *Boana xerophylla*. (**B**) *Rhinella marina*. (**C**) *Phyllomedusa bicolor*. Photo credits: A and B by Guilherme Melo-dos-Santos; C by Anne Grace A. C. Marques.

**Figure 2 microorganisms-13-01610-f002:**
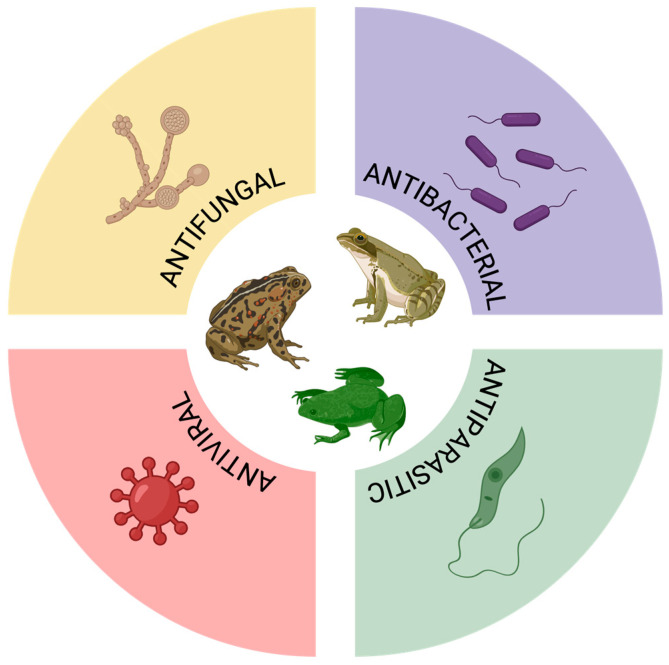
Antimicrobial properties of anuran toxins.

**Table 1 microorganisms-13-01610-t001:** Anuran antimicrobial peptides with antibacterial activity.

Peptides	Species	Target	Ref.
Alyteserin-1Ma, -1Mb, -2Ma	* Alytes maurus *	*S. aureus* and *E. coli*	[27]
Alyteserin-2Mb	* Alytes maurus *	* S. aureus *	[27]
Alyteserin-1a, -1b, -1c, -2a	* Alytes obstetricans *	*S. aureus* and *E. coli*	[28]
Brevinin-1CG1	* Amolops chunganensis *	*B. licheniformis*, *R. rhodochrous*, and *E. coli*	[29]
Brevinin-1CG2, -1CG3	* Amolops chunganensis *	*S. aureus*, *S. carnosus*, *E. faecalis*, *B. licheniformis*, *R. rhodochrous*, *P. faecalis*, *S. rubidaea*, and *E. coli*	[29]
Brevinin-1CG4	* Amolops chunganensis *	*S. aureus*, *S. carnosus*, *E. faecalis*, *B. licheniformis*, *R. rhodochrous*, *P. faecalis*, *S. rubidaea*, *P. aeruginosa* and *E. coli*	[29]
Brevinin-1CG5	* Amolops chunganensis *	*S. aureus*, *S. carnosus*, *E. faecalis*, *B. licheniformis*, *R. rhodochrous*, *P. faecalis*, *S. rubidaea*, *K. pneumoniae*, *P. aeruginosa* and *E. coli*	[29]
Brevinin-2CG1	* Amolops chunganensis *	*S. aureus*, *S. carnosus*, *B. licheniformis*, *R. rhodochrous*, *P. faecalis*, *S. rubidaea*, and *E. coli*	[29]
Esculentin-2CG1, Palustrin-2CG1	* Amolops chunganensis *	*S. aureus*, *S. carnosus*, *E. faecalis*, *B. licheniformis*, *R. rhodochrous*, *P. faecalis*, *S. rubidaea*, *K. pneumoniae* and *E. coli*	[29]
Temporin-CG1	* Amolops chunganensis *	*S. aureus*, *S. carnosus*, *E. faecalis*, *B. licheniformis*, *R. rhodochrous*, *P. faecalis* and *S. rubidaea*	[29]
Temporin-CG2, -CG3	* Amolops chunganensis *	*S. aureus*, *S. carnosus*, *E. faecalis*, *B. licheniformis*, *R. rhodochrous* and *S. rubidaea*	[29]
Jindongenin-1a	* Amolops jingdongensis *	*S. aureus*, *E. faecalis*, *E. cloacae*, *B. pyocyaneus*, *S. dysenteriae*, *K. pneumoniae*, *P. aeruginosa* and *E. coli*	[30]
Palustrin-2AJ1	* Amolops jingdongensis *	*E. faecalis*, *B. pyocyaneus*, *S. dysenteriae*, *K. pneumoniae*, *P. aeruginosa* and *E. coli*	[30]
Brevinin-1RTa	* Amolops ricketti *	*S. aureus*, *B. licheniformis*, *R. rhodochrous* and* P. faecalis*	[31]
Brevinin-1RTb, -2RTb	* Amolops ricketti *	*S. aureus*, *S. carnosus*, *B. licheniformis*, *R. rhodochrous* and* P. faecalis*	[31]
Brevinin-2RTa	* Amolops ricketti *	*S. aureus*, *S. carnosus*, *B. licheniformis*, *R. rhodochrous*, *P. faecalis* and* S. rubidaea*	[31]
Ascaphin-1	* Ascaphus truei *	*E. cloacae*, *K. pneumoniae* and* E. coli*	[32]
Ascaphin-3	* Ascaphus truei *	* E. coli *	[32]
Ascaphin-5	* Ascaphus truei *	*S. aureus*, *S. epidermidis*, *E. faecalis*, *E. cloacae*, *K. pneumoniae*, *P. aeruginosa* and *E. coli*	[32]
Ascaphin-7	* Ascaphus truei *	*S. aureus* and *E. coli*	[32]
Ascaphin-8	* Ascaphus truei *	*S. aureus*, *S. epidermidis*, *E. cloacae*, *K. pneumoniae*, *P. aeruginosa* and *E. coli*	[32]
Pleurain-A1, -A2	* Babina pleuraden *	*S. aureus*, *H. pylori*, *S. dysenteriae* and *E. coli*	[33]
Maximin 1, Maximin 2, Maximin 3, Maximin 4, Maximin 5	* Bombina maxima *	*S. aureus*, *B. megaterium*, *B. pyocyaneus*, *S. dysenteriae*, *K. pneumoniae*, and *E. coli*	[34]
Maximin H1, Maximin H2, Maximin H3, Maximin H4	* Bombina maxima *	*S. aureus*, *B. pyocyaneus* and *E. coli*	[34]
Maximin S4	* Bombina maxima *	*M. humenis* and* U. urealyticum*	[35]
Bombinin H1, Bombinin H3, Bombinin H4	* Bombina variegata *	*S. aureus* and* E. coli*	[36]
Tigerinin-RC1, -RC2	* Fejervarya cancrivora *	*S. aureus*, *B. subtilis*, *P. aeruginosa* and *E. coli*	[37]
Hylaseptin P1	* Hyla punctata *	*S. aureus*, *P. aeruginosa* and *E. coli*	[38]
Hylain 1, Hylain 2	* Hyla simplex *	*S. aureus*, *B. cereus*, *B. subtilis*, *S. dysenteriae*, *P. aeruginosa* and *E. coli*	[39]
Temporin GHa, Temporin GHb, Temporin GHc, Temporin GHd	* Hylarana guentheri *	*S. aureus*, MRSA, *B. subtilis*, *V. alginolyticus*, *P. aeruginosa* and *E. coli*	[40]
Brevinin-2LTa, -2LTb	* Hylarana latouchii *	*S. aureus*, *S. carnosus*, *B. licheniformis*, *R. rhodochrous*, *P. faecalis*, *S. rubidaea*, *P. aeruginosa* and *E. coli*	[41]
Brevinin-2LTc	* Hylarana latouchii *	*S. aureus*, *R. rhodochrous* and* P. faecalis*	[41]
Esculentin-1LTa	* Hylarana latouchii *	*S. aureus*, *S. carnosus*, *B licheniformis*, *R. rhodochrous*, *P. faecalis*, *S. rubidaea* and *E. coli*	[41]
Esculentin-2LTa	* Hylarana latouchii *	*S. aureus*, *S. carnosus*, *B. licheniformis*, *R. rhodochrous* and *P. faecalis*	[41]
Palustrin-2LTa	* Hylarana latouchii *	*S. aureus*, *B. licheniformis* and* P. faecalis*	[41]
Temporin-LTe	* Hylarana latouchii *	*S. aureus*, *S. carnosus*, *B. licheniformis* and* R. rhodochrous*	[41]
Temporin PTa	* Hylarana picturata *	*MRSA*, *B. subtilis* and* E. coli*	[42]
Dermaseptin-L1	* Hylomantis lemur *	* E. coli *	[43]
Phylloseptin-L1	* Hylomantis lemur *	* S. aureus *	[43]
Hylin a1	* Hypsiboas albopunctatus *	*S. aureus*, *B. subtilis*, *E. faecalis*, *P. aeruginosa* and *E. coli*	[44]
Raniseptin-1	* Hypsiboas raniceps *	*S. aureus*, *P. aeruginosa* and *E. coli*	[45]
Kassinatuerin-2Ma, Kassorin S	* Kassina maculata *	* S. aureus *	[46,47]
Kassinatuerin-1	* Kassina senegalensis *	*S. aureus* and *E. coli*	[48,49]
Kassorin S	* Kassina senegalensis *	* S. aureus *	[47]
Fallaxin	* Leptodactylus fallax *	*E. cloacae*, *P. mirabilis*, *K. pneumoniae*, *P. aeruginosa* and *E. coli*	[50]
Laticeptin	* Leptodactylus laticeps *	*E. cloacae*, *K. pneumoniae*, *P. aeruginosa* and* E. coli*	[51]
Pentadactylin	* Leptodactylus pentadactylus *	*S. aureus*, *S. epidermidis*, *E. faecalis*, *S.* group B, *E. cloacae*, *K. pneumoniae*, *P. aeruginosa* and* E. coli*	[52]
Ocellatin 1, Ocellatin 2, Ocellatin 3	* Leptodactylus ocellatus *	* E. coli *	[53]
Ocellatin 4	* Leptodactylus ocellatus *	*S. aureus* and *E. coli*	[54]
Brevinvin	* Limnonectes fujianensi *	*S. aureus* and* E. coli*	[55]
Brevinin-1VLa, -1VLc	* Lithobates vaillanti *	*S. aureus* and* E. coli*	[56]
Brevinin-IVLd, -1VLe	* Lithobates vaillanti *	* S. aureus *	[56]
Ranatuerin-2VLb	* Lithobates vaillanti *	* E. coli *	[56]
Aurein 2.1	* Litoria aurea *	*S. epidermidis*, *B. cereus*, *M. luteus*, *S. uberis*, *L. lactis* and *L. innocua*	[57]
Aurein 2.2, Aurein 2.4	* Litoria aurea *	*S. aureus*, *S. epidermidis*, *B. cereus*, *M. luteus*, *S. uberis*, *L. lactis* and *L. innocua*	[57]
Aurein 2.3	* Litoria aurea *	*S. aureus*, *S. epidermidis*, *B. cereus*, *M. luteus*, *L. lactis* and *L. innocua*	[57]
Aurein 2.5	* Litoria aurea *	*S. aureus*, MRSA, *S. epidermidis*, *B. cereus*, *M. luteus*, *E. faecalis*, Vancomicyn-resistant *E. faecalis*, *L. lactis*, *L. innocua*, *A. baumannii*, *K. pneumoniae*, *P. aeruginosa* and *E. coli*	[57,58]
Aurein 3.1, Aurein 3.2	* Litoria aurea *	*S. aureus*, *S. epidermidis*, *M. luteus*, *S. uberis*, *L. lactis* and *L. innocua*	[57]
Aurein 5.2	* Litoria aurea *	*S. uberis* and* L. lactis*	[57]
Citropin 1.1	* Litoria citropa *	*B. cereus*, *L. lactis*, *L. innocua*, *M. luteus*, *S. aureus*, *S. epidermidis* and* S. uberis*	[59]
Caerin 1.1, Uperin 3.5	* Litoria ewingi *	*B. cereus*, *L. lactis*, *L. innocua*, *M. luteus*, *P. multocida*, *S. aureus*, *S. epidermidis* and *S. uberis*	[60]
Fallaxidin 3.1, Fallaxidin 3.2	* Litoria fallax *	*E. faecalis* and* L. lactis*	[61]
Fallaxidin 4.1	* Litoria fallax *	*S. epidermidis*, *S. uberis*, *M. luteus* and* L. lactis*	[61]
Caerin 1.1	* Litoria splendida *	*B. cereus*, *L. lactis*, *L. innocua*, *M. luteus*, *P. multocida*, *S. aureus*, *S. epidermidis* and *S. uberis*	[62]
Aurein 1.1	* Litoria raniformis *	*B. cereus*, *S. uberis*, *L. lactis* and *L. innocua*	[57]
Aurein 1.2	* Litoria raniformis *	*S. aureus*, *S. epidermidis*, *B. cereus*, *M. luteus*, *S. uberis*, *L. lactis*, *P. multocida* and *L. innocua*	[57]
Aurein 2.1	* Litoria raniformis *	*S. epidermidis*, *B. cereus*, *M. luteus*, *S. uberis*, *L. lactis* and *L. innocua*	[57]
Aurein 2.5, Aurein 2.6	* Litoria raniformis *	*S. aureus*, *S. epidermidis*, *B. cereus*, *M. luteus*, *L. lactis* and *L. innocua*	[57]
Aurein 3.1, Aurein 3.2	* Litoria raniformis *	*S. aureus*, *S. epidermidis*, *Micrococus. luteus*, *S. uberis*, *L. lactis* and *L. innocua*	[57]
Aurein 3.3	* Litoria raniformis *	*S. aureus*, *S. epidermidis*, *M. luteus*, *S. uberis* and* L. lactis*	[57]
Aurein 5.2	* Litoria raniformis *	*S. uberis* and* L. lactis*	[57]
Caerin 1.1, Citropin 1.1, Citropin 1.2	* Litoria subglandulosa *	*S. aureus*, *S. epidermidis*, *M. luteus*, *B. cereus*, *E. faecalis*, *S. uberis*, *L. lactis* and *L. innocua*	[63]
Megin 1, 2	* Megophrys minor *	*S. aureus*, *B. subtilis*, *S. dysenteriae*, and *E. coli*	[64]
Japonicin-1Npa	* Nanorana parkeri *	* S. aureus *	[65]
Japonicin-1Npb	* Nanorana parkeri *	*S. aureus* and *N. asteroides*	[65]
Parkerin	* Nanorana parkeri *	*S. aureus*, *E. faecium* and *A. baumannii*	[65]
Brevinin-1HN1, Brevinin-1V	* Odorrana hainanensis *	*S. aureus*, *S. carnosus*, *B. licheniformis*, *E. faecium*, *E. faecalis*, *R. rhodochrous*, *P. faecalis* and* E. coli*	[66]
Brevinin-2HS2, Temporin-HN1	* Odorrana hainanensis *	*S. aureus*, *S. carnosus*, *B. licheniformis*, *E. faecium*, *R. rhodochrous* and* P. faecalis*	[66]
Temporin-HN2	* Odorrana hainanensis *	*S. aureus*, *S. carnosus*, *B. licheniformis*, *R. rhodochrous* and* P. faecalis*	[66]
Temporin-SHa	* Pelophylax saharicus *	*S. aureus*, *E. faecalis*, *B. megaterium*, *P. aeruginosa* and *E. coli*	[67]
Temporin-SHb	* Pelophylax saharicus *	*S. aureus*, *B. megaterium* and *E. coli*	[67]
Temporin-SHc	* Pelophylax saharicus *	*S. aureus* and *B. megaterium*	[67]
Temporin-SHd	* Pelophylax saharicus *	*S. aureus*, *E. faecalis*, *B. megaterium*, *L. ivanovii*, *A. baumannii*, *K. pneumoniae* and *E. coli*	[68,69]
Temporin-SHe	* Pelophylax saharicus *	*S. aureus*, *E. faecalis*, *B. megaterium*, *L. ivanovii*, *A. baumannii*, *K. pneumoniae*, *P. aeruginosa*, *S. enteritidis* and *E. coli*	[69]
Temporin-SHf	* Pelophylax saharicus *	*S. aureus*, *E. faecalis*, *B. megaterium* and *E. coli*	[70]
Phylloxin	* Phyllomedusa bicolor *	*B. megaterium*, *M. luteus*, *C. glutamicum*, *R. meliloti*, and *E. coli*	[71]
Distinctin	* Phyllomedusa distincta *	*S. aureus*, *E. faecalis*, *P. aeruginosa* and *E. coli*	[72]
Phylloseptin-1	* Phyllomedusa hypochondrialis *	*S. aureus*, *E. faecalis*, *P. aeruginosa* and *E. coli*	[73]
Dermaseptin B2	* Phyllomedusa * * sauvagei *	*S. aureus*, *S. haemolyticus*, *B. subtilis*, *B. megaterium*, *Listeria monocytogenes*, *Salmonella enterica* serovar *Typhimurium, K. pneumoniae* and* E. coli*	[74]
Dermaseptin S9	* Phyllomedusa * * sauvagei *	*S. haemolyticus*, *B. subtilis*, *B. megaterium*, *L. monocytogenes*, *S. typhimurium*, *K. pneumoniae* and* E. coli*	[74]
Phylloseptin-1	* Phyllomedusa * * sauvagei *	*S. aureus* and* E. coli*	[75]
Pseudin-2	* Pseudis paradoxa *	*S. aureus*, *S. epidermidis*, *B. subtilis*, *B. megaterium*, *L. monocytogenes*, *S. typhimurium*, *E. cloacae*, *K. pneumoniae*, *P. aeruginosa* and* E. coli*	[76]
Esculentin-1Ara, -1ARb, Ranatuerin-2ARb, Temporin-1AR	* Rana areolata *	*S. aureus* and* E. coli*	[77]
Palustrin-2AR, -3AR, Ranatuerin-2Ara	* Rana areolata *	* E. coli *	[77]
Brevinin-1AVa	* Rana arvalis *	* L. lactis *	[78]
Brevinin-1AUa	* Rana aurora aurora *	*S. aureus*, *S. epidermidis*, *E. faecalis*, *S.* group B, *E. cloacae*, *K. pneumoniae*, *P. aeruginosa* and* E. coli*	[79]
Brevinin-1AUb	* Rana aurora aurora *	*S. aureus*, *S. epidermidis*, *E. faecalis*, *S.* group B, *E. cloacae*, *P. aeruginosa* and* E. coli*	[79]
Ranatuerin-2AUa	* Rana aurora aurora *	*S. aureus*, *S. epidermidis*, *S.* group B, *E. cloacae*, *K. pneumoniae*, *P. aeruginosa* and* E. coli*	[79]
Brevinin-1Ba, -1Bc	* Rana berlandieri *	* S. aureus *	[80]
Brevinin-1Bb, -1Bd, -1Be, -1Bf, Esculentin-2B, Ranatuerin-2B	* Rana berlandieri *	*S. aureus* and* E. coli*	[80]
Ranatuerin-1	* Rana catesbeiana *	*S. aureus* and* E. coli*	[81]
Ranatuerin-2, -3, -4, -6, -7, -8, -9	* Rana catesbeiana *	* S. aureus *	[81]
Japonicin-2CHa	* Rana chaochiaoensis *	* S. aureus *	[82]
Brevinin-1CEa, Temporin-1CEa, -1CEb	* Rana chensinensis *	*S. aureus*, *B. cereus*, *Lactococcus lactis* and *E. coli*	[83]
Brevinin-2CE, Palustrin-2CE, Temporin-1Cee	* Rana chensinensis *	*S. aureus*, *Bacillus subtilis*, *P. aeruginosa and E. coli*	[84]
Chensinin-1CEb	* Rana chensinensis *	*S. aureus*, *B. subtilis* and *E. coli*	[84]
Chensinin-3CE	* Rana chensinensis *	*S. aureus* and *E. coli*	[84]
Temporin-1Ceh	* Rana chensinensis *	*S. aureus*, MRSA, *E. faecalis*, *K. pneumoniae*, *P. aeruginosa* and *E. coli*	[85]
Ranalexin-1Ca, Ranatuerin-1C, -2Cb	* Rana clamitans *	*S. aureus* and *E. coli*	[86]
Temporin-1Cb, -1Cd, -1Ce	* Rana clamitans *	* S. aureus *	[86]
Brevinin-1CDYa, Dybowskin-1CDYa, -2CDYa	* Rana dybowskii *	*S. aureus* and *E. coli*	[87]
Japonicin-1CDYa	* Rana dybowskii *	* E. coli *	[87]
Brevinin-1E, -2E, Esculentin-1, -2a	* Rana esculenta *	*S. aureus*, *B. megaterium*, *P. aeruginosa* and *E. coli*	[88]
Ranacyclin E	* Rana esculenta *	*Staphylococcus lentus*, *M. luteus*, *B. megaterium*, *Y. pseudotuberculosis* and *P. syringae pv tobaci*	[89]
Esculentin-1Gra, Brevinin-2GRa, -2GRb, -1GRa, -2GRc, Nigrocin-2GRb	* Rana grahami *	*S. aureus* and *E. coli*	[90]
Nigrocin-2GRa, -2GRc	* Rana grahami *	* E. coli *	[90]
Ranatuerin-1Ga, -2G, Ranalexin-1G	* Rana grylio *	*S. aureus* and *E. coli*	[91]
Temporin-1Gb, -1Gc, -1Gd	* Rana grylio *	* S. aureus *	[91]
Japonicin-1	* Rana japonica *	* E. coli *	[92]
Japonicin-2	* Rana japonica *	*S. aureus* and *E. coli*	[92]
Brevinin-1Lb, Esculentin-2L, Ranatuerin-2La, -2Lb, Temporin-1La	* Rana luteiventris *	*S. aureus* and *E. coli*	[80]
Temporin-1Lb, -1Lc	* Rana luteiventris *	* S. aureus *	[80]
Nigrocin-1, -2	* Rana nigromaculata *	*M. luteus*, *S. dysentariae*, *K. pneumoniae*, *S*. *typhimurium* and *P. aeruginosa*	[93]
Brevinin-1Oka, -1OKc, Ranatuerin-2OK	* Rana okinavana *	*S. aureus* and* E. coli*	[94]
Brevinin-1PLb, -1PLc, Esculentin-1Pla, -1PLb, -2Pla, Palustrin-3a	* Rana palustris *	*S. aureus* and* E. coli*	[95]
Ranatuerin-2PLb, -2PLc, -2PLd, -2PLf, Palustrin-1b, -2b, -2c, -3b	* Rana palustris *	* E. coli *	[95]
Temporin-1PLa	* Rana palustris *	* S. aureus *	[95]
Brevinin-Ipa, -1Pb, -1Pc, -1Pd	* Rana pipiens *	*S. aureus* and* E. coli*	[80]
Esculentin-2P	* Rana pipiens *	* E. coli *	[80]
pLR	* Rana pipiens *	*S. lentus*, *M. luteus* and *B. megaterium*	[89]
Ranatuerin-2P	* Rana pipiens *	*S. aureus* and* E. coli*	[80]
Ranatuerin-2Pb	* Rana pipiens *	*S. aureus*, MRSA, and *E. coli*	[96]
Temporin-1P	* Rana pipiens *	* S. aureus *	[80]
Brevinin-1PRa	* Rana pirica *	* S. aureus *	[97]
Brevinin-2Pra, -2PRb, -2PRc, -2PRd, -2PRe	* Rana pirica *	*S. aureus*, *E. cloacae*, *K. pneumoniae*, *P. aeruginosa* and *E. coli*	[97]
Ranatuerin-2Pra	* Rana pirica *	* E. coli *	[97]
Gaegurin-1	* Rana rugosa *	*S. epidermidis*, *B. subtilis*, *M. luteus*, *S. dysentariae*, *K. pneumoniae*, *P. putida*, *P. aeruginosa* and *E. coli*	[98]
Gaegurin-2, -3, -4, -5	* Rana rugosa *	*S. epidermidis*, *B. subtilis*, *M. luteus*, *S. dysentariae*, *K. pneumoniae*, *S*. *typhimurium*, *P. putida*, *P. aeruginosa* and *E. coli*	[98]
Gaegurin-6	* Rana rugosa *	*S. epidermidis*, *B. subtilis*, *M. luteus*, *S. dysentariae*, *K. pneumoniae*, *P. putida*, *P. aeruginosa* and *E. coli*	[98]
Rugosin A	* Rana rugosa *	*S. aureus*, *B. subtilis*, *M. luteus*, *S. pyogenes* and *E. coli*	[99]
Rugosin B	* Rana rugosa *	*S. aureus*, *B. subtilis*, *M. luteus*, *S. pyogenes*, *P. aeruginosa* and *E. coli*	[99]
Brevinin-2SKa, Peptide VR-23	* Rana sakuraii *	*S. aureus* and *E. coli*	[100]
Brevinin-2SKb, Ranatuerin-2SKa	* Rana sakuraii *	* E. coli *	[100]
Temporin-1SKa, -1SKb	* Rana sakuraii *	* S. aureus *	[100]
Brevinin-1Spa, -1SPb, -1SPd, Temporin-1SPb	* Rana septentrionalis *	*S. aureus* and *E. coli*	[101]
Melittin-related peptide	* Rana tagoi *	*S. aureus* and *E. coli*	[102]
Ranacyclin T	* Rana temporaria *	*S. lentus*, *M. luteus*, *B. megaterium*, *Y. pseudotuberculosis*, *P. syringae pv tobaci* and *E. coli*	[89]
Temporin A	* Rana temporaria *	*S. aureus*, *S. epidermidis*, *Staphylococcus capitis*, *B. megaterium*, *Y. pseudotuberculosis*, *S. pyogenes*, *E. coli*, *E. faecium*, *E. faecalis*, *S. maltophilia*, *A. baumannii*, *P. syringae pv tobaci* and *P. aeruginosa*	[103,104,105]
Temporin B	* Rana temporaria *	*S. aureus*, *B. megaterium*, *Y. pseudotuberculosis*, *S. pyogenes*, *E. coli*, *E. faecium*, *S. maltophilia*, *A. baumannii* and *P. aeruginosa*	[103,104]
Temporin G	* Rana temporaria *	*S. aureus*, *E. faecium*,* S. maltophilia*, *A. baumannii* and *P. aeruginosa*	[104]
Temporin L	* Rana temporaria *	*S. aureus*, MRSA, *S. epidermidis*, *Staphylococcus capitis*, *E. faecalis*, *B. megaterium*, Vancomicyn-resistant *E. faecalis*, *S. pyogenes*, *A. baumannii*, *K. pneumoniae*, *P. syringae pv tobaci*, *Y. pseudotuberculosis*, *P. aeruginosa* and *E. coli*	[58,105]
Tigerinin 1, 2, 3	* Rana tigerina *	*S. aureus*, *B. subtilis*, *M. luteus*, *P. putida* and *E. coli*	[106]
Tigerinin 4	* Rana tigerina *	*S. aureus*, *P. putida* and *E. coli*	[106]
Brevinin-1TSa	* Rana tsushimensis *	*S. aureus*, MRSA, *S. epidermidis*, *E. faecalis*, *S.* group B, *E. cloacae*, *P. aeruginosa* and *E. coli*	[107]
Brevinin-2TSa	* Rana tsushimensis *	*S. aureus*, MRSA, *S. epidermidis*, *E. faecalis*, *E. cloacae*, *K. pneumoniae*, *P. aeruginosa* and *E. coli*	[107]
Temporin-1Va	* Rana virgatipes *	*S. aureus*, *S. epidermidis*, *E. faecalis*, *S.* group B,* K. pneumoniae*, *E. cloacae*, *P. aeruginosa* and* E. coli*	[108]
Temporin-1Vb	* Rana virgatipes *	*S. aureus*, *S. epidermidis*, *E. faecalis* and* S.* group B	[108]
Temporin-1Vc	* Rana virgatipes *	*S. aureus*, *S. epidermidis*, *E. faecalis*, *S.* group B,* K. pneumoniae*, *E. cloacae* and* P. aeruginosa*	[108]
Brevinin-2GHk	* Sylvirana guentheri *	*S. aureus* and* E. faecalis*	[109]
Uperin 2.1	* Uperoleia rnjobergii *	*S. uberis*, *L. mesenteroides* and *L. innocua*	[110]
Uperin 2.8	* Uperoleia rnjobergii *	*S. epidermidis*, *B. cereus*, *S. uberis* and* L. lactis*	[110]
Uperin 3.1	* Uperoleia rnjobergii *	*S. epidermidis*, *M. luteus*, *S. uberis* and* L. lactis*	[110]
Uperin 3.5	* Uperoleia rnjobergii *	*S. aureus*, *S. epidermidis*, *B. cereus*, *M. luteus*, *S. uberis*, *L. lactis*, *P. multocida* and *L. innocua*	[110]
Uperin 3.6	* Uperoleia rnjobergii *	*S. aureus*, *S. epidermidis*, *B. cereus*, *M. luteus*, *S. uberis*, *L. lactis* and *L. innocua*	[110]
CPF-AM1, -AM4, PGLa-AM1	* Xenopus amieti *	*S. aureus* and* E. coli*	[111]
Magainin-AM2	* Xenopus amieti *	* E. coli *	[111]
CPF-SP1, -SP2, -SP3	* Xenopus andrei *	*S. aureus*, *A. baumannii*, *K. pneumoniae*, *P. aeruginosa* and* E. coli*	[112]
Magainin-AN2	* Xenopus andrei *	*K. pneumoniae*, *P. aeruginosa* and* E. coli*	[112]
PGLa-AN2, XPF-AN1	* Xenopus andrei *	*S. aureus* and* E. coli*	[112]
PGLa-SP1, XPF-SP1, -SP2	* Xenopus andrei *	*A. baumannii* and* E. coli*	[112]
CPF-B1, PGLa-B2, XPF-B2	* Xenopus borealis *	*S. aureus* and* E. coli*	[113]
Magainin-B2	* Xenopus borealis *	* E. coli *	[113]
CPF-AM1	* Xenopus clivii *	*A. baumannii*, *K. pneumoniae* and *P. aeruginosa*	[114]
CPF-C1	* Xenopus clivii *	*S. aureus*, *A. baumannii*, *K. pneumoniae*, *P. aeruginosa* and* E. coli*	[114]
CPF-C2	* Xenopus clivii *	*S. aureus* and* E. coli*	[114]
Magainin-C1, -C2, XPF-C1	* Xenopus clivii *	* E. coli *	[114]
Peptide XT-7	* Xenopus clivii *	*A. baumannii*, *K. pneumoniae* and *P. aeruginosa*	[114]
Magainin 1	* Xenopus laevis *	*S. aureus*, *S. epidermidis*, *K. pneumoniae*, *C. freundii*, *E. cloacae*, *S. marcescens*, *P. putida*, *P. aeruginosa* and *E. coli*	[115]
Magainin 2	* Xenopus laevis *	*S. aureus*, *S. epidermidis*, *S. pyogenes*, *K. pneumoniae*, *C. freundii*, *E. cloacae*, *S. marcescens*, *P. putida*, *P. aeruginosa* and *E. coli*	[115,116]
PGLa, XPF	* Xenopus laevis *	*S. aureus*, *S. pyogenes*, *P. aeruginosa* and *E. coli*	[116]
CPF-M1, -MW1, -MW2, Magainin-M1, PGLa-MW1	* Xenopus muelleri *	*S. aureus* and *E. coli*	[117]
Magainin-MW1	* Xenopus muelleri *	* E. coli *	[117]
XT-1, -3, -6, -7	* Xenopus tropicalis *	*S. aureus* and *E. coli*	[118]
XT-2, -4	* Xenopus tropicalis *	* E. coli *	[118]

Abbreviations: *Acinetobacter baumannii* (*A. baumannii*); *Bacillus cereus* (*B. cereus*); *Bacillus licheniformis* (*B. licheniformis*); *Bacillus megaterium* (*B. megaterium*); *Bacillus pyocyaneus* (*B. pyocyaneus*); *Bacillus subtilis* (*B. subtilis*); *Citrobacter freundii* (*C. freundii*); *Corynebacterium glutamicum* (*C. glutamicum*); *Enterobacter cloacae* (*E. cloacae*); *Enterococcus faecalis* (*E. faecalis*); *Escherichia coli* (*E. coli*); *Helicobacter pylori* (*H. pylori*); *Klebsiella pneumoniae* (*K. pneumoniae*); *Leuconostoc mesenteroides* (*L. mesenteroides*); *Leuconostoc lactis* (*L. lactis*); *Listeria innocua* (*L. innocua*); *Listeria ivanovii* (*L. ivanovvii*); *Listeria monocytogenes* (*L. monocytogenes*); *methicillin-resistant Staphylococcus aureus* (*MRSA*); *Micrococcus luteus* (*M. luteus*); *Mycoplasma humenis* (*M;. humenis*); *Nocardia asteroides* (*N. asteroides*); *Pasteurella multocida* (*P. multocida*); *Proteus mirabilis* (*P. mirabilis*); *Pseudomonas aeruginosa* (*P. aeruginosa*); *Pseudomonas putida* (*P. putida*); *Pseudomonas syringae pv tobaci* (*P. syringae pv tobaci*); *Psychrobacter faecalis* (*P. faecalis*); *Rhizobium meliloti* (*R. meliloti*); *Rhodococcus rhodochrous* (*R. rhodochrous*); *Salmonella enterica serotype Enteritidis* (*S. enteritidis*); *Salmonella enterica serovar Typhimurium* (*S. typhimurium*); *Serratia marcescens* (*S. marcescens*); *Serratia rubidaea* (*S. rubidaea*); *Shigella dysenteriae* (*S. dysenteriae*); *Staphylococcus aureus* (*S. aureus*), *Staphylococcus carnosus* (*S. carnosus*); *Staphylococcus epidermidis* (*S. epidermidis*); *Staphylococcus haemolyticus* (*S. haemolyticus*); *Staphylococcus lentus* (*S. lentus*); *Streptococcus group B* (*S. group B*); *Streptococcus pyogenes* (*S. pyogenes*); *Streptococcus uberis* (*S. uberis*); *Ureaplasma urealyticum* (*U. urealyticum*); *Vibrio alginolyticus* (*V. alginolyticus*); *Yersinia pseudotuberculosis* (*Y. pseudotuberculosis*).

**Table 2 microorganisms-13-01610-t002:** Anuran antifungal toxins.

Toxin	Species	Target	Ref.
Arenobufagin, Gamabufotalin, Telocinobufagin	* Anaxyrus boreas *	* Batrachochytrium dendrobatidis *	[122]
LC6, LC7	* Leptophryne cruentata *	* Trichophyton mentagrophytes *	[123]
RJ7, RJ8	* Rhacophorus margaritifer *	* Trichophyton mentagrophytes *	[123]
Dehydrobufotenine (alkaloid)	* Bufo arenarum *	* Alternaria solani *	[124]
Compound 16h (Dehydrobufotenine anolog)	* Bufo arenarum *	*Alternaria solani* *Rhizoctonia solani* *Botrytis cinereal* *Cercospora arachidicola*	[124]
Compound 16c (Dehydrobufotenine anolog)	* Bufo arenarum *	* Alternaria solani * * Rhizoctonia solani *	[124]
Compound 16d (Dehydrobufotenine anolog)	* Bufo arenarum *	*Sclerotinia sclerotiorum* *Alternaria solani* *Rhizoctonia solani* *Botrytis cinereal* *Cercospora arachidicola*	[124]
Compound 16j (Dehydrobufotenine anolog)	* Bufo arenarum *	*Sclerotinia sclerotiorum* *Alternaria solani* *Botrytis cinereal* *Cercospora arachidicola*	[124]
Compound 19 (Dehydrobufotenine anolog)	* Bufo arenarum *	* Alternaria solani * * Cercospora arachidicola *	[124]
Esculentin-1 (Esc-1a(1-21)NH_2_ and Esc-1b(1-18)NH_2_)	* Pelophylac lessonae/ridibundus *	* Candida albicans *	[125]
B2RP, Nigroain, Esculetin-2	*Hylarana erythraea*	* Candida albicans *	[126,127]
Brevinin-1 and 2	*Hylarana nigrovittatta*	* Candida albicans *	[128]

**Table 3 microorganisms-13-01610-t003:** Antiviral activity of anuran toxins.

Toxin	Species	Target	Ref.
AR-23	* Rana tagoi *	HSV-1, MeV, HPIV-2, HCoV-229E, SARS-CoV-2, BoHV-1, CpHV-1, CDV, BVDV e SBV	[139]
Brevinin-1	* Rana brevipoda porsa *	HSV-1 e HSV-2	[142]
Brevinin-2	* Rana septentrionalis *	HIV-1	[141]
Brevinin-2GHk	* Fejervarya limnocharis *	ZIKV	[148]
Caerin 1.1	* Litoria splendida *	HIV and HPV	[130]
Caerin 1.10	* Litoria splendida *	HIV	[131]
Caerin 1.19	* Litoria gracilenta *	HIV	[131]
Caerin 1.2	* Litoria caerula *	HIV	[131]
Caerin 1.20	Hybrid between the female *Litoria splendida* and the male *Litoria caerulea*	HIV	[131]
Caerin 1.3	* Litoria caerula *	HIV	[131]
Caerin 1.4	* Litoria caerula *	HIV	[131]
Caerin 1.5	* Litoria caerula *	HIV	[131]
Caerin 1.6	* Litoria xanthomera *	HIV	[131]
Caerin 1.7	* Litoria xanthomera *	HIV	[131]
Caerin 1.9	* Litoria chloris *	HIV and HPV	[130]
Dahlein 5.6	* Litoria dahlii *	HIV	[130]
Dermaseptin-S1	* Phyllomedusa sauvagii *	HIV, FV3, CCV e HSV-1	[133,134,138]
Dermaseptin-S3	* Phyllomedusa sauvagii *	RABV	[135]
Dermaseptin-S4	* Phyllomedusa sauvagii *	HSV-1, HSV-2, ZIKV, HIV e RABV	[133,134,135,136,137]
Dermaseptin-S9	* Phyllomedusa sauvagei *	HIV-1 e SARS-CoV-2	[134]
Esculentin-1ARb	* Rana areolata *	HIV	[130]
Esculentin-1GN	* Hylarana guentheri *	H5N1, H1N1, DENV 2 and 3	[134,149]
Esculentin-2P	*Rana luteiventris*, *Rana berlandieri* and *Rana pipiens*	HIV	[130]
Figainin 2	* Boana raniceps *	CHIKV, YFV, and DENV 4	[150]
Frenatin 2	* Litoria infrafrenata *	YFV	[132]
Frenatin 2.3S	* Sphaenorhynchus lacteus *	YFV	[132]
Hylin a1	* Hypsiboas albopunctatus *	BoHV-1, CpHV-1, CDV, BVDV e SBV HSV-1 e HSV-2	[139]
Maculatin 1.1	*Litoria genimaculate* and *Litoria eucnemis*	HIV and HPV	[130,134]
Magainin 1	* Xenopus laevis *	HSV-1 e HSV-2	[147]
Magainin 2	* Xenopus laevis *	HSV-1 e HSV-2	[147]
Maximin 1	* Bombina maxima *	HIV-1	[34]
Maximin 3	* Bombina maxima *	HIV	[34]
Maximin 4	* Bombina maxima *	HIV	[34]
Maximin 5	* Bombina maxima *	HIV	[34]
Palustrin-3AR	* Rana areolata *	HIV	[130]
Ranatuerin 2P	*Rana pipiens* and* Rana pretiosa*	HIV	[130]
Ranatuerin 6	* Rana catesbeiana *	HIV	[130]
Ranatuerin 9	* Rana catesbeiana *	HIV	[130]
RV-23	* Rana draytonii *	BoHV-1, CpHV-1, CDV, BVDV e SBV	[139]
Temporin A	* Rana temporaria *	CCV, FV3	[145]
Temporin B	* Rana temporaria *	HSV-1	[143,144]
Temporin G	* Rana temporaria *	IAV e VPH	[144,151]
Temporin L	* Rana temporaria *	HSV-1, HSV-2 paramixovírus, IAV and SARS-CoV-2	[144,145]
Temporin-SHa	* Pelophylax saharica *	HSV-1	[144]
Uperin 3.6	* Uperoleia inundata *	HIV	[130]
Urumin	* Hydrophylax bahuvistara *	H1N1	[134,146]
Yodha	* Indosylvirana aurantiaca *	ZIKV, DENV1, DENV2, DENV3 e DENV4	[134,140]

Abbreviations: bovine herpesvirus type 1 (BoHV-1); Bovine viral diarrhea virus (BVDV); canine distemper virus (CDV); caprine herpesvirus type 1 (CpHV-1); Chikungunya virus (CHIKV); channel catfish virus (CCV); dengue virus serotypes 1; 2; 3; and 4 (DENV-1; DENV-2; DENV-3; and DENV-4); frog virus 3 (FV3); human coronavirus 229E (HCoV-229E); human immunodeficiency virus (HIV); human immunodeficiency virus type 1 (HIV-1); human papillomavirus (HPV); human parainfluenza virus type 2 (HPIV-2); herpes simplex virus types 1 and 2 (HSV-1 and HSV-2); influenza A virus (IAV); influenza A virus subtype H1N1 (H1N1); influenza A virus subtype H5N1 (H5N1); measles virus (MeV); rabies virus (RABV); Schmallenberg virus (SBV); severe acute respiratory syndrome coronavirus 2 (SARS-CoV-2); yellow fever virus (YFV); and Zika virus (ZIKV).

**Table 4 microorganisms-13-01610-t004:** Antiparasitic activity of anuran toxins.

Toxin	Species	Target	Ref.
Bombinin H2	* Bombina variegata *	*L. donovani*; *L. pifanoi*	[155]
Bombinin H4	* Bombina variegata *	*L. donovani*; *L. pifanoi*	[155]
Dermaseptin 1	* Phyllomedusa nordestina *	* Trypanosoma cruzi *	[156]
Dermaseptin 4	* Phyllomedusa nordestina *	* Trypanosoma cruzi *	[156]
DRS-H10	* Phyllomedusa nordestina *	*Leishmania amazonensis*; *Leishmania infantum*	[157]
Ds01	* Phyllomedusa oreades *	*Trypanosoma cruzi*; *Schistosoma mansoni*	[158,159]
Figainin 1	* Boana raniceps *	* Trypanosoma cruzi *	[160]
Hellebrigenin	* Rhinella jimi *	*Leishmania (L.) chagasi*; *Trypanosoma cruzi*	[161]
Phylloseptins PS-4	* Phyllomedusa oreades *	* Trypanosoma cruzi *	[73]
Phylloseptins PS-5	* Phyllomedusa oreades *	* Trypanosoma cruzi *	[73]
Phylloseptin-1 (PSN-1)	*Phyllomedusa azurea*	*Leishmania amazonensis*	[162]
Phylloseptin 7	* Phyllomedusa nordestina *	*Trypanosoma cruzi*, *Leishmania (L.) infantum*	[156]
Phylloseptin 8	* Phyllomedusa nordestina *	* Trypanosoma cruzi *	[156]
SaFr1	* Siphonops annulatus *	*Trypanosoma cruzi*, *Leishmania (L.) infantum*	[163]
Telocinobufagin	* Rhinella jimi *	* Leishmania (L.) chagasi *	[161]
Temporin-SHd	* Pelophylax saharicus *	* Leishmania infantum *	[68]
Temporin-SHe	* Pelophylax saharicus *	* Leishmania infantum *	[69]
Temporin-1Sa	* Pelophylax (Rana) saharica *	* Leishmania infantum *	[164]

## Data Availability

No new data were created or analyzed in this study.

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
