# Peer review of "Investigating the Antimicrobial Activity of Anuran Toxins"

_microorganisms, 2025, doi:10.3390/microorganisms13071610_

Round 1

Reviewer 1 Report

Comments and Suggestions for Authors

Manuela B. Pucca, et al., submitted their manuscript entitled "Investigating the Antimicrobial Activity of Anuran Toxins."
Below are my comments, which I hope will be helpful to the authors:
Overall, I consider the topic relevant to the field of research.
To make the manuscript more robust, I recommend:
1. Section 2 can be considerably reduced. This information has already been extensively described in other articles. It could focus on explaining the species considered in the present work and their importance.
2. Section 3 can also be reduced since the topic of the article is antimicrobial peptides. Other anuran secretions can be described, but the importance of the antimicrobial peptides isolated and evaluated should be indicated.
3. Section 5, Advanced Research, should be significantly modified. As the authors point out, the number of in vitro studies on antimicrobial peptides is extensive; However, an analysis can be added of what needs to be done so that those peptides with greater potential can advance to in vivo studies or even the clinical phase and achieve new therapeutic alternatives.

Author Response

Manuela B. Pucca, et al., submitted their manuscript entitled "Investigating the Antimicrobial Activity of Anuran Toxins."

Below are my comments, which I hope will be helpful to the authors:

Overall, I consider the topic relevant to the field of research.

To make the manuscript more robust, I recommend:

  1. Section 2 can be considerably reduced. This information has already been extensively described in other articles. It could focus on explaining the species considered in the present work and their importance.

Response: Thanks for the comments to enhance the manuscript quality. The Section 2 was reduced, focusing on the anuran species with antimicrobial toxins and the second paragraph about anuran respiration was removed.

  1. Section 3 can also be reduced since the topic of the article is antimicrobial peptides. Other anuran secretions can be described, but the importance of the antimicrobial peptides isolated and evaluated should be indicated.

Response: Thank you for the comments. Following your suggestion, section 3 was reduced.

  1. Section 5, Advanced Research, should be significantly modified. As the authors point out, the number of in vitro studies on antimicrobial peptides is extensive; However, an analysis can be added of what needs to be done so that those peptides with greater potential can advance to in vivo studies or even the clinical phase and achieve new therapeutic alternatives.

Response: Thank you very much for your suggestion. The section 5 was modified accordingly.

Reviewer 2 Report

Comments and Suggestions for Authors

Though the review topic is interesting and important in both fundamental and application study of anuran toxins, the mamuscript presented here is not well organized and interpreted in terms of the structure and storytelling. This reviewer understands the topic is might be too big to be included in a relatively short review paper. Some suggestions are listed for authors to improve their paper.

1) The authors can choose one of two well-studied anuran toxins (or toxin family) as examples to explain the machanism to explain the signicance of the anuran toxins.

2) Most of the toxins need struture indicates if they are available in database or published literature.

3) The organization needs significant improvement while balancing the contents of the whole manuscript.

4) For conclusions, the authors also need to include more perpectives from their understanding and share some useful insignts with researchers and readers in this field. The current one is not acceptable. The authors also need to explain why this review is important and what are differences of this paper from other similar ones. 

Author Response

Though the review topic is interesting and important in both fundamental and application study of anuran toxins, the manuscript presented here is not well organized and interpreted in terms of the structure and storytelling. This reviewer understands the topic is might be too big to be included in a relatively short review paper. Some suggestions are listed for authors to improve their paper.

1) The authors can choose one of two well-studied anuran toxins (or toxin family) as examples to explain the machanism to explain the signicance of the anuran toxins.

Response: We respectfully decided, by consensus among all authors, to retain the current format of the manuscript. While we appreciate the reviewer’s thoughtful suggestion to highlight one or two well-studied anuran toxins as illustrative examples, we believe that presenting the data as originally structured offers a more comprehensive and integrative overview. This broader approach allows us to better capture the diversity, mechanisms, and potential significance of anuran toxins within the scope of the review. Nonetheless, we thank the reviewer for the valuable input, which helped us re-evaluate and reaffirm our presentation strategy.

2) Most of the toxins need struture indicates if they are available in database or published literature.

Response: We chose not to include structural data, as our manuscript focuses on the functional aspects of anuran toxins rather than their structural characteristics..

3) The organization needs significant improvement while balancing the contents of the whole manuscript.

Response: We sincerely thank the reviewer for this valuable observation. We have carefully revised the manuscript to improve its overall organization and ensure a more balanced presentation of the content throughout. We have reorganized the sections to follow a clearer and more logical progression of ideas, we adjusted each section to achieve better balance, avoiding overemphasis on specific parts, we ensured smoother transitions between paragraphs and sections to enhance readability. Also, we refined the introduction and conclusion to better frame the main objectives and contributions of the work. We believe these changes have significantly improved the clarity and coherence of the manuscript.

4) For conclusions, the authors also need to include more perpectives from their understanding and share some useful insignts with researchers and readers in this field. The current one is not acceptable. The authors also need to explain why this review is important and what are differences of this paper from other similar ones.

Response: Thank you for your suggestion. The conclusion section was improved accordingly. 

Reviewer 3 Report

Comments and Suggestions for Authors

Dear Authors,

The manuscript “Investigating the Antimicrobial Activity of Anuran Toxins” by Pucca et al should undergo the necessary revisions before being accepted for further publication steps.
(1) Please add a “Materials and Methods” section and include the number of articles cited and the slogans used by the authors in selecting the Articles used to write this Review.
2. lines 65-66 - a map with continents with the distribution of anurans would be more attractive.
3. subsections 2.1, 2.2, 2.3 - you might want to add references to Fig 1 A, B, C, respectively
4. line 116 - “prevent”..... no continuation of the sentence
5. description of alkaloids should be improved, as quanidins and lipophiles act as toxins, are a defense for anurans against predators, and indoles show a number of positive properties and have pharmacological potential. In addition, two different mechanisms of action that are worth considering to make Figures, in order to diversify Manuscript.
6. biogenic amines - it is worth adding a few sentences on their mechanism of action (consistently to alkaloids ).
7. line 206 - “immune defense” - please explain how this happens

8. table 1 - please organize into Gram-negative, Gram-positive and others by adding columns. This will improve the readability of this huge table. You can also divide it into 3 smaller ones: separately for Gram-negative....Also, the authors should use the full generic names of bacteria here, since this is their first appearance. They can be written in abbreviation (then consistently all of them), but under the table there should be a legend explaining each abbreviation used, e.g. P. aeruginosa - Pseudomonas aeruginosa. 
9. from line 226 the citation is in italics - please correct it
10. table 2 - for readability, please separate yeast-like fungi from mold-like fungi by adding columns
11. line 289 “compounds 6, 16c, 16d, 16h, 16j and 19” - add to the description what characterized these compounds, what they had in common. The authors of the cited work for the Manuscript gave numbers, but here without specific names this notation is inconsistent.
12. line 312 - the names of viruses at 1 use should necessarily include the full development of abbreviations. This also applies to the other abbreviations. Please apply to the entire Manuscript. 
13. table 3 - under the table, please include a legend with an explanation of the abbreviations used in it
14. line 373 - “cancer treatment” - what cancer? 
15. to Chapter 5, the authors should add information on the studies of the compounds described here in neurological disorders and those related to muscle tone.

16 In addition, each chapter describing the activity should include a subsection discussing the mechanism of action of these compounds on a particular cell: bacterial, fungal, viral or parasitic. Please add this information.

In addition, in some places there is a need to correct the wording of some sentences because they are unclear, such as line 194/195
In addition, when using generic names (applies to bacteria and fungi), the authors should consider using the full name only for 1 use, for the next use the abbreviation should be used, e.g. I - Candida albicans, II - C. albicans

Kind regards

Comments on the Quality of English Language

In some places there is a need to correct the wording of some sentences because they are unclear, such as line 194/195

Author Response

Dear Authors,

The manuscript “Investigating the Antimicrobial Activity of Anuran Toxins” by Pucca et al should undergo the necessary revisions before being accepted for further publication steps.
(1) Please add a “Materials and Methods” section and include the number of articles cited and the slogans used by the authors in selecting the Articles used to write this Review.

Response: Thank you for all the comments aimed at improving the quality of the manuscript. The intention of this work was to provide a narrative literature review, rather than a systematic review; therefore, the authors prefer not to include a Materials and Methods section. However, we conducted searches using terms such as antibacterial anurans, antifungal anurans, and antiviral anurans across several databases, including PubMed, ScienceDirect, and Google Scholar, in order to gather the most comprehensive information possible on antimicrobial toxins from anurans.

2. lines 65-66 - a map with continents with the distribution of anurans would be more attractive.

Response: We appreciate the reviewer’s insightful suggestion regarding the inclusion of a distribution map of anurans across continents, as it would indeed enhance the visual appeal and contextual understanding of the manuscript. After careful consideration, we have decided not to include this map in the current version. Our rationale is that the primary focus of the manuscript is on the biochemical and pharmacological properties of anuran-derived antimicrobial compounds rather than on biogeographical distribution per se. Additionally, including such a map would require substantial space and resources that may detract from the core scientific content and discussion.

3. subsections 2.1, 2.2, 2.3 - you might want to add references to Fig 1 A, B, C, respectively

Response: Figures were provided by the authors and credits are in the legend.

4. line 116 - “prevent”..... no continuation of the sentence

Response: The sentence was completed.

5. description of alkaloids should be improved, as quanidins and lipophiles act as toxins, are a defense for anurans against predators, and indoles show a number of positive properties and have pharmacological potential. In addition, two different mechanisms of action that are worth considering to make Figures, in order to diversify Manuscript.

Response: We thank the reviewer for the valuable comments regarding the description of alkaloids and their diverse biological properties. We agree that compounds such as guanidines, lipophilic toxins, and indole alkaloids possess important pharmacological and ecological roles, including defense mechanisms and a wide range of bioactivities. However, we would like to clarify that the primary focus of our review is on antimicrobial molecules derived from anuran secretions. Therefore, our intention in this section was to briefly summarize the main classes of alkaloids present in anuran toxins, rather than provide an in-depth discussion of their broader pharmacological effects or ecological functions. The descriptions were kept concise to maintain the manuscript's central theme and avoid shifting the focus away from antimicrobial relevance. Regarding the suggestion to include figures illustrating different mechanisms of action, we appreciate this idea. However, considering that the antimicrobial mechanisms of most anuran-derived alkaloids are not yet fully characterized, and given the review’s scope, we opted to refrain from adding such figures at this stage. Nonetheless, we acknowledge the value of this insight and we hope this clarification is acceptable and appreciate the reviewer’s understanding of the manuscript’s thematic focus.

6. biogenic amines - it is worth adding a few sentences on their mechanism of action (consistently to alkaloids ).

Response: The referred sentence was added.

7. line 206 - “immune defense” - please explain how this happens.

Response: To clarify the mechanism of action related to “immune defense,” the following sentence was added: “These peptides are synthesized and secreted mainly in the skin, acting as a first line of protection against pathogens through their cytolytic activity.”

8. table 1 - please organize into Gram-negative, Gram-positive and others by adding columns. This will improve the readability of this huge table. You can also divide it into 3 smaller ones: separately for Gram-negative....Also, the authors should use the full generic names of bacteria here, since this is their first appearance. They can be written in abbreviation (then consistently all of them), but under the table there should be a legend explaining each abbreviation used, e.g. P. aeruginosa - Pseudomonas aeruginosa. 

Response: To improve the readability of Table 1, all bacterial names were abbreviated, and a legend explaining each abbreviation was added below the table. However, we chose not to divide the table into separate ones for Gram-positive and Gram-negative bacteria, as this would require significantly more space.

9. from line 226 the citation is in italics - please correct it

Response: It was corrected.

10. table 2 - for readability, please separate yeast-like fungi from mold-like fungi by adding columns.

Response: Thank you for the suggestion. However, we prefer not to separate yeast-like fungi from mold-like fungi into different columns, as our intention was to present all fungal species affected by the peptides in a unified and comprehensive manner. Only Candida albicans is classified as a yeast-like fungus; all the other species listed in the table are mold-like fungi. Since the antimicrobial activity discussed in the table refers broadly to fungi, we believe that a combined presentation maintains clarity without affecting the scientific value of the data.

11. line 289 “compounds 6, 16c, 16d, 16h, 16j and 19” - add to the description what characterized these compounds, what they had in common. The authors of the cited work for the Manuscript gave numbers, but here without specific names this notation is inconsistent.

Response: The text was revised to refer to “six of the analogs of dehydrobufotenine” instead of listing the compound numbers. This approach was chosen to avoid excessive length in the manuscript, as providing detailed descriptions of how each analog was designed and synthesized would be too extensive.

12. line 312 - the names of viruses at 1 use should necessarily include the full development of abbreviations. This also applies to the other abbreviations. Please apply to the entire Manuscript. 

Response: We have included the full names of all viruses at first mention.

13. table 3 - under the table, please include a legend with an explanation of the abbreviations used in it.

Response: The abbreviations used in Table 3 have already been explained in a legend provided below the table.

14. line 373 - “cancer treatment” - what cancer? 

Response: The types of cancer targeted by the treatments are specified in the text.

15. to Chapter 5, the authors should add information on the studies of the compounds described here in neurological disorders and those related to muscle tone.

Response: Thank you for your thoughtful suggestion. We acknowledge the relevance of addressing potential applications of anuran compounds in neurological disorders and conditions related to muscle tone. However, after reviewing the scope and objectives of the present manuscript, the authors agreed to maintain the current focus on the antimicrobial, antifungal, and antiviral properties of these compounds, as they represent the most consistently studied and evidenced biological activities within the available literature. While certain anuran alkaloids and peptides have shown neuromodulatory or myotropic effects in experimental contexts, these findings remain limited, and comprehensive data regarding their therapeutic application in neurological or neuromuscular disorders are still scarce and fragmented. For this reason, we opted not to include a dedicated discussion on this topic in Chapter 5 to preserve the coherence and focus of the review. Nonetheless, we recognize the importance of this line of research and suggest it as a promising area for future investigation.

16 In addition, each chapter describing the activity should include a subsection discussing the mechanism of action of these compounds on a particular cell: bacterial, fungal, viral or parasitic. Please add this information.

Response: A paragraph was added to section 4. Antimicrobial Compounds, including information on the main mechanisms of action of AMPs in general.

In addition, in some places there is a need to correct the wording of some sentences because they are unclear, such as line 194/195.

Response: Section 3.4, titled Peptide, has been revised to enhance clarity for readers.

In addition, when using generic names (applies to bacteria and fungi), the authors should consider using the full name only for 1 use, for the next use the abbreviation should be used, e.g. I - Candida albicans, II - C. albicans

Response: Abbreviations for bacterial and fungal names have been consistently used after their initial full mention.

Round 2

Reviewer 2 Report

Comments and Suggestions for Authors

No further comments on this revised manucsript.

Author Response

Dear reviewer,

thank you for your time to improve our manuscript.

Best regards,